# Oxidative Stress and Antioxidant Defense in the Brain of Bat Species with Different Feeding Habits

**DOI:** 10.3390/ijms241512162

**Published:** 2023-07-29

**Authors:** Pabulo Henrique Rampelotto, Nikolas Raphael Oliveira Giannakos, Diego Antonio Mena Canata, Francielly Dias Pereira, Fernanda Schäfer Hackenhaar, María João Ramos Pereira, Mara Silveira Benfato

**Affiliations:** 1Bioinformatics and Biostatistics Core Facility, Instituto de Ciências Básicas da Saúde, Universidade Federal do Rio Grande do Sul, Porto Alegre 91501-970, Brazil; prampelotto@hcpa.edu.br; 2Graduate Program in Biological Sciences, Pharmacology and Therapeutics, Universidade Federal do Rio Grande do Sul, Porto Alegre 91501-970, Brazil; 3Biophysics Department, Universidade Federal do Rio Grande do Sul, Porto Alegre 91501-970, Brazil; 4Graduate Program in Cellular and Molecular Biology, Universidade Federal do Rio Grande do Sul, Porto Alegre 91501-970, Brazil; 5Department of Medical Biosciences, Umeå University, 901 87 Umeå, Sweden; 6Graduate Program in Animal Biology, Universidade Federal do Rio Grande do Sul, Porto Alegre 91501-970, Brazil

**Keywords:** oxidative stress, brain function, brain damage, frugivorous, nectarivorous, insectivorous, hematophagous

## Abstract

Assessing the levels of oxidative stress markers and antioxidant enzymes in the brain is crucial in evaluating its antioxidant capacity and understanding the influence of various dietary patterns on brain well-being. This study aimed to investigate the antioxidant status and oxidative damage in the brain of bat species with different feeding habits to gain insights into their protective mechanisms against oxidative stress and their interspecific variation. The levels of oxidative damage markers and the activities of antioxidants were measured in the brain of four bat species with different feeding habits, namely insectivorous, frugivorous, nectarivorous, and hematophagous. Insectivorous bats showed higher levels of SOD and fumarase compared to the other groups, while hematophagous bats showed lower levels of these enzymes. On the other hand, the activities of glutathione peroxidase and glutathione S-transferase were higher in hematophagous bats and lower in insectivorous bats. The carbonyl groups and malondialdehyde levels were lower in frugivores, while they were similar in the other feeding guilds. Nitrite and nitrate levels were higher in the hematophagous group and relatively lower in all other groups. The GSSG/GSH ratio was higher in the hematophagous group and lower in frugivores. Overall, our results indicate that the levels of oxidative stress markers and the activities of antioxidant enzymes in the brain vary significantly among bat species with different feeding habitats. The findings suggest that the antioxidant status of the brain is influenced by diet and feeding habits.

## 1. Introduction

The brain is a vital organ particularly susceptible to oxidative damage due to its elevated oxygen consumption, abundance of phospholipids and polyunsaturated fatty acids (which are highly vulnerable to oxidants), presence of redox-active metals, and relatively low levels of antioxidant enzymes [1]. Antioxidant enzymes and oxidative stress markers are important indicators of the antioxidant status of the brain and can provide insight into the impact of different feeding habits on brain health [2].

The impact of different feeding habits on brain oxidative stress is an area of research that sheds light on the relationship between nutrition and brain health. Recent studies suggest that certain dietary patterns can influence the level of oxidative stress in the brain. For instance, consuming a diet high in fats and excessive sugar has been associated with an increased risk of cognitive problems and neurological dysfunction [3]. Conversely, foods rich in vitamins, minerals, and antioxidants are known to nourish the brain and protect it from oxidative stress [4].

Bats are a diverse group of mammals with different feeding habits ranging from frugivorous to insectivorous and hematophagous groups [5]. Feeding habits are one of the factors that can influence the brain of bats, potentially shaping their cognitive abilities and adaptive strategies, though the specific impact of different feeding habits may vary [6]. Previous studies have shown that feeding habits can affect the antioxidant status of different bat tissues [7], but the impact of feeding habits on the antioxidant status of the brain remains poorly understood. 

Therefore, the aim of this study was to investigate the oxidative stress and antioxidant status in the brain of four bat species with different feeding habits by measuring the levels of oxidative damage markers (carbonyl groups and malondialdehyde), antioxidant enzymes (superoxide dismutase, fumarase, glutathione peroxidase, and glutathione S-transferase), and non-enzymatic antioxidants (nitrites and nitrates as well as the GSSG/GSH ratio). 

Understanding how different feeding habits impact oxidative stress and antioxidant defense in bat brains can shed light on the physiological adaptations that have allowed bats to thrive in diverse ecological niches. Furthermore, the implications of our study extend beyond the scope of bat biology and provide insights into the broader field of comparative oxidative physiology. By elucidating the interplay between feeding habits, antioxidant status, and oxidative damage in bat brains, our findings may have implications for understanding the evolutionary adaptations of other organisms with diverse feeding habits and their responses to oxidative stress.

## 2. Results

Figure 1 shows the oxidative damage in the brain tissue of the four bat feeding habits as measured using two oxidative damage markers. The carbonyl groups and malondialdehyde levels were lower in frugivores, while they were similar in the other feeding habit groups. 

Figure 2 presents the activity of antioxidant enzymes measured in the brain tissue of the four bat species. Insectivorous bats showed higher levels of SOD and fumarase compared to the other groups, while hematophagous bats showed lower levels of these enzymes. On the other hand, the activities of glutathione peroxidase and glutathione S-transferase were higher in hematophagous bats and lower in insectivorous bats. These results suggest that the antioxidant defense mechanisms in the brain of bats are influenced by their dietary habits and feeding guilds and may reflect adaptations to different levels of oxidative stress imposed by their diets.

Figure 3 shows the levels of non-enzymatic antioxidants in the brains of the same bats. Nitrite and nitrate levels were higher in the hematophagous group and relatively lower in all other groups. The GSSG/GSH ratio was higher in the hematophagous group and lower in frugivores. The results suggest that the oxidative damage in the brain tissue of the different bat guilds varies according to their feeding habits.

To compare the overall oxidative damage and antioxidant status among the four bat species, a PCA was conducted with the oxidative biomarkers and the antioxidant enzymes (Figure 4). The results of the PCA revealed a clear differentiation in the clustering of samples based on their feeding habits. It also showed that the first two components explained 84.6% of the variation. 

These results were confirmed by the pairwise PERMANOVA test among samples grouped according to bat species (Table 1).

The correlation between the antioxidants and oxidative markers measured in the brain of each bat species is presented in Figure 5. While MDA was negatively correlated with carbonyl in the nectarivorous group, fumarase was negatively correlated with the GSSG/GSH ratio in the frugivorous group. In the insectivorous group, SOD was negatively correlated with fumarase and GPx. In the hematophagous group, carbonyl was negatively correlated with MDA and GPx, while GST was positively correlated with nitrites and nitrates (NO_2_&NO_3_).

The overall correlation between the antioxidants and oxidative markers measured in the brains of bats is presented in Figure 6. While SOD and the GSSG/GSH ratio were positively correlated (*p* = 0.005), they were inversely correlated with the other enzymes and oxidative markers. Fumarase, glutathione peroxidase, and glutathione S-transferase were highly correlated with each other and with carbonyl groups and malondialdehyde (*p* < 0.01). On the other hand, the levels of nitrites and nitrates did not present any significant correlation with enzymes (*p* > 0.05).

## 3. Discussion

The brain is a vital organ that is particularly vulnerable to oxidative stress, which can lead to cellular damage and dysfunction. In this study, we have accessed the antioxidant status of the brain in four bat species with different feeding habits by measuring the levels of oxidative damage markers and the activity of enzymatic and non-enzymatic antioxidants. By measuring the oxidative damage and antioxidants in the brain of different bat species, we have gained new insights into the protective mechanisms against oxidative stress and their interspecific variation. This information can help us understand how different diets and feeding habits impact the antioxidant defense systems of the brain. Additionally, as bats have diverse feeding habits and exhibit a range of metabolic adaptations, they represent a useful model for studying nutritional neuroscience and oxidative stress in mammals.

It is worth noting that while studying wildlife is challenging for controlling confounding factors, we were careful in trying to control major factors that are common interferences in wildlife studies. All the collected animal species in our study exhibit nocturnal habits and follow similar circadian cycles. They share similar foraging times, including feeding and flight activities. Moreover, we observed that the entry and exit times of the caves were practically identical for all species. The capturing process took place at the beginning of the night to ensure that all bats had not consumed any food and were in the same basal physiological state. In addition, none of these four species engage in hibernation or torpor, which are physiological adaptations that allow animals to conserve energy during periods of unfavorable conditions. Based on our observations, we believe that the differences in these habits among the species are not significant enough to account for the variations we observed and analyzed in terms of redox metabolism. Instead, we attribute these differences to variations in the diet of these species. Furthermore, the stress associated with the capture of the animals, including the time spent in the net, handling of the individuals, and euthanasia procedures, was consistent among all four species. These activities were carried out by the same individuals, ensuring uniformity in the stressors applied. Consequently, any biases that could have arisen from these procedures would have been equivalent for all species, minimizing the potential for significant intraspecies differences.

Our results showed that the activity of antioxidant enzymes in the brain of bats varies depending on their diet and feeding habits. In general, SOD and fumarase were higher in the insectivorous and lower in the hematophagous groups, while GPx and GST presented an opposite status, i.e., higher in hematophagous and lower in insectivorous. These results suggest that the antioxidant defense mechanisms in the brain of bats are influenced by their feeding habits. Insectivorous bats exhibited elevated levels of SOD and fumarase, which may be linked to their protein-rich diet, primarily composed of insects. Interestingly, recent studies have shown that a high-protein diet induces oxidative stress in the brain in animal models [8,9]. The oxidative stress triggered by a high-protein diet could also interact with biological molecules to disrupt the normal synthesis and repair of DNA, leading to DNA damage [10], which can be measured using the levels of cytosolic fumarase [11].

In contrast, the lower activity of SOD and fumarase in hematophagous bats may indicate a reduced need for antioxidant protection due to their diet primarily being composed of blood. The main blood component that could lead to oxidative stress would be Fe. However, in normal physiological situations, Fe input to the brain is regulated by the blood–brain barrier [12]. On the other hand, the higher activity of GPx and GST in hematophagous bats may suggest a higher demand for the detoxification of reactive oxygen species produced during the digestion of blood components [13]. In addition, the intermediate and similar levels of these enzymes in frugivorous and nectarivorous bats may be explained by some common characteristics of their diets. Both frugivorous and nectarivorous bats feed on fruits and nectar, which are rich sources of antioxidants such as vitamin C and polyphenols, compounds known to pass through the blood–brain barrier [14,15]. These antioxidants can scavenge free radicals and reduce oxidative stress, thereby reducing the need for high levels of antioxidant enzymes. Additionally, both diets may have lower levels of pro-oxidants, such as heme from animal products, compared to insectivorous or hematophagous diets. This observation may potentially contribute to the explanation of the lower levels of antioxidant enzymes found in these bat species. However, further research is needed to confirm these hypotheses.

Regarding the observed oxidative damage in the brain, the levels of carbonyl groups and malondialdehyde were usually lower in frugivores, while similar in the other bats. Nitrite and nitrate levels were higher in the hematophagous group and relatively lower in all other groups. The GSSG/GSH ratio was higher in the hematophagous group and lower in frugivores. The results suggest that the oxidative damage in the brain varies depending on the feeding habits of bats. The lower levels of carbonyl groups and malondialdehyde in frugivorous bats may indicate a lower level of oxidative stress and/or a more efficient antioxidant system in these bats compared to the other feeding guilds. These lower levels in frugivorous bats could be explained by the fact that fruits, which are the primary food of frugivorous bats, are rich in antioxidants and other phytochemicals that could help protect against oxidative damage [16]. Antioxidants neutralize free radicals, preventing them from damaging cellular components such as proteins and lipids, which can result in the production of carbonyl groups and malondialdehyde. Additionally, fruits are generally low in polyunsaturated fatty acids, which are more susceptible to oxidative damage and can result in higher levels of malondialdehyde [17]. Therefore, the lower levels of oxidative damage observed in frugivorous bats could be due to their consumption of a diet rich in antioxidants and low in polyunsaturated fatty acids. 

On the other hand, the higher levels of nitrites and nitrates in the brain of hematophagous bats may be due to the presence of these compounds in their diet. Blood contains high levels of nitrate, which is converted to nitrite by bacteria in the oral cavity of these bats [18]. Nitrite can then be converted to nitric oxide (NO) through the action of nitrite reductase enzymes. Nitric oxide is a signaling molecule that plays an essential role in the regulation of various physiological processes in the brain, including vasodilation, neurotransmitter release, and synaptic plasticity [19].

However, excessive nitric oxide production can lead to the formation of reactive nitrogen species (RNS), which can cause oxidative damage to brain tissues [20]. Hematophagous bats may be more susceptible to RNS-mediated oxidative stress due to the higher levels of nitrites and nitrates in their diet. Additionally, the high levels of iron in blood may promote the formation of highly reactive nitrogen species such as peroxynitrite, which can cause significant damage to brain tissues [21]. Therefore, while nitric oxide plays an essential role in the regulation of brain functions, excessive production of RNS can be harmful. The higher levels of nitrites and nitrates in the brain of hematophagous bats may reflect the potential for increased RNS production, which could have implications for their brain function and health.

The higher GSSG/GSH ratio in hematophagous bats may indicate an imbalance in the redox state of the cells, which can lead to increased oxidative damage. The GSSG/GSH ratio is an indicator of the oxidative stress status in cells, as it reflects the balance between oxidized (GSSG) and reduced (GSH) forms of glutathione, which is an important intracellular antioxidant [22]. A higher GSSG/GSH ratio indicates an increase in oxidative stress levels, as it suggests a higher rate of glutathione oxidation to counteract the oxidative damage caused by reactive oxygen species (ROS).

Hematophagous bats may experience a higher level of oxidative stress than frugivorous bats, leading to the higher GSSG/GSH ratio observed in our study. On the other hand, frugivorous bats feed on fruits and nectar, which are known to be rich in antioxidants such as vitamins C and E, carotenoids, and polyphenols [16]. These compounds may provide protection against oxidative stress by scavenging ROS and preventing their formation. Therefore, frugivorous bats may experience a lower level of oxidative stress than hematophagous bats, leading to the lower GSSG/GSH ratio observed in our study.

A graphical scheme summarizing our biochemical findings in each bat group is presented in Figure 7.

When the correlation between the antioxidants and oxidative markers measured in the brain was calculated for each bat species, a distinct pattern was observed for each group, which likely reflects the specific metabolic demands, oxidative stress profiles, and adaptive responses associated with each feeding strategy.

In the nectarivorous group, MDA was negatively correlated with carbonyl. The negative correlation suggests that as the levels of lipid peroxidation (MDA) decrease, the levels of protein oxidation (carbonyl) increase in nectarivorous bats. This finding might indicate a compensatory mechanism where increased protein oxidation occurs as a response to reduced lipid peroxidation. The underlying reason for this correlation could be the different feeding habits and metabolic demands of nectarivorous bats, which may lead to variations in the generation of reactive oxygen species (ROS) and subsequent oxidative damage.

In the frugivorous group, fumarase was negatively correlated with the GSSG/GSH ratio. This correlation may imply that reduced fumarase activity affects the redox balance in frugivorous bats, leading to an alteration in the GSSG/GSH ratio. It is possible that the frugivorous diet and associated metabolic processes influence the activity of fumarase and glutathione metabolism, resulting in this observed correlation.

In the insectivorous group, SOD was negatively correlated with fumarase and GPx. The negative correlation indicates that as SOD activity decreases, the activities of fumarase and GPx increase in insectivorous bats. This correlation suggests a potential compensatory mechanism where decreased SOD activity is associated with enhanced fumarase and GPx activities. The reason behind this correlation could be the higher metabolic rate and increased production of ROS in insectivorous bats, which necessitates higher antioxidant defense capacities.

In the hematophagous group, carbonyl was negatively correlated with MDA and GPx, while GST was positively correlated with nitrites and nitrates (NO_2_&NO_3_). The negative correlation between carbonyl and MDA suggests that increased protein oxidation is associated with reduced lipid peroxidation in hematophagous bats. Additionally, the negative correlation between carbonyl and GPx indicates that reduced protein oxidation is linked to increased GPx activity. On the other hand, the positive correlation between GST and NO_2_&NO_3_ implies that higher GST levels are associated with increased levels of these nitrogen species. These correlations suggest that hematophagous bats exhibit a different oxidative stress and antioxidant defense pattern compared to the other feeding groups. The reasons for these specific correlations in hematophagous bats could be related to their unique feeding behavior and associated metabolic processes, which may result in distinct oxidative stress profiles and antioxidant defense mechanisms.

Identifying the overall correlation pattern among the enzymatic and other biochemical parameters also presented some interesting results. SOD, fumarase, GPx, and GST all play important roles in protecting the brain against oxidative damage [23]. However, they have different functions and may be affected differently by various factors [24]. As such, it is possible that these enzymes are involved in different pathways or mechanisms that are affected by the different diets and feeding habits of bats. For example, SOD is an antioxidant enzyme that catalyzes the dismutation of superoxide radicals into oxygen and hydrogen peroxide, while GPx and GST are also antioxidant enzymes that protect against oxidative stress by reducing hydrogen peroxide and lipid peroxides [23]. Cytosolic fumarase is a part of the DNA damage response that is recruited from the cytosol to the nucleus upon DNA damage induction [25]. The inverse correlation between SOD-fumarase and GPx-GST thus suggests a trade-off between energy metabolism and antioxidant defense in the brain [26]. In other words, the brain may need to balance the energy demand and the protection against oxidative stress, and changes in one pathway may affect the other. In addition, the inverse observed correlation between the levels of SOD and GPx in the brain may be due to the way these enzymes are regulated. SOD is regulated primarily at the transcriptional level, whereas GPx is regulated at the post-transcriptional level [27,28]. This means that changes in the availability of transcription factors can increase or decrease the levels of SOD in the brain. On the other hand, changes in the availability of reduced glutathione can modulate the activity of GPx. Therefore, if the brain experiences an increase in oxidative stress, the levels of SOD may increase rapidly to scavenge superoxide radicals. This increase in SOD levels may cause a temporary decrease in the levels of reduced glutathione, which could decrease the activity of GPx. As a result, there may be an inverse correlation between the levels of SOD and GPx in the brain during periods of high oxidative stress [29]. 

Nitrites and nitrates did not show significant correlations with the enzymes in the study, likely because they are not directly involved in the antioxidant system. Nitrites and nitrates are instead end products of the metabolism of nitric oxide (NO), a signaling molecule that plays various roles in physiological processes, including vasodilation, neurotransmission, and immune response [19]. NO can be metabolized into nitrites and nitrates through various pathways, and their levels can serve as biomarkers for NO metabolism. However, the antioxidant enzymes studied in this research are mainly involved in the removal of ROS, not NO, and their activities may not directly affect nitrite and nitrate levels. 

To our knowledge, this is the first study to investigate the oxidative stress and antioxidant defense in the brains of bat species with different feeding habits. One limitation of our study is the relatively small sample size of animals per group. A larger sample size would have provided a more robust analysis and increased the generalizability of our findings. However, obtaining a larger sample size was logistically difficult given the challenges of capturing and studying wild animals. In addition, Brazilian law is very restrictive on the capturing of wildlife for research purposes, especially considering threatened species like bats, and an authorized license was only obtained for the planned experimental design of using 10 animals per group. To overcome this limitation, a rigorous statistical analysis was employed. Furthermore, we were able to control major factors that are common interferences in wildlife studies and could have impacted our results, e.g., species, sex, and physiological state. For these reasons, the results obtained are consistent and meaningful, providing new insights into how different diets and feeding habits impact the antioxidant defense systems of the brain. These findings not only shed light on the intricate relationship between diet and brain health but also provide valuable information regarding the adaptive strategies and potential cognitive disparities among bat species. 

Through the exploration of oxidative metabolism and antioxidant defense mechanisms, this study uncovers novel perspectives that enhance our understanding of how diet and feeding habits shape the brain’s antioxidant defense systems.

## 4. Materials and Methods

### 4.1. Ethical Aspects

This study received ethical approval from the Animal Use Ethics Committee of the Federal University of Rio Grande do Sul (CEUA/UFRGS, Approval No. 28645). Furthermore, all procedures conducted in this research adhered to the Brazilian law for the scientific use of animals (Law No. 11,794) and the Guidelines of the National Council for the Control of Animal Experimentation (CONCEA). The collection of zoological material was carried out under a license authorized by the SISBIO Biodiversity Information and Authorization System (No. 47202-1), the National System for the Management of Genetic Heritage and Associated Traditional Knowledge (SisGen), and the National Council for the Control of Animal Experimentation (No. 33339).

### 4.2. Animal and Sample Collection

A total of 39 adult male bats were captured in southern Brazil between the summer of 2018 and the winter of 2019. The capturing methods utilized were dip nets, mist nets, or harp traps, depending on the specific shelter type (Table 2). The captured bat species included *Glossophaga soricina* (*n* = 10), *Sturnira lilium* (*n* = 10), *Molossus molossus* (*n* = 10), and *Desmodus rotundus* (*n* = 9). The capturing process took place at the beginning of the night to ensure that all bats had not consumed any food. Subsequently, euthanasia was performed on the animals using intraperitoneal injection of a combination of xylazine (10 mg/kg) and ketamine (60 mg/kg) before removing all organs. The harvested organs were immediately frozen in liquid nitrogen and stored at −80 °C for subsequent analysis and testing.

### 4.3. Organ Processing

The brains were manually macerated by Potter in a solution of 10 mL potassium phosphate buffer (30 mmol/L), KCL (120 mmol/L), PMSF (0.201 mmol/L), and desferroxamine (1.5 mmol/L). Subsequently, the samples were sonicated (3×) for 10 s and centrifuged at 1700× *g* (2×) for 10 min, and the final supernatant was aliquoted into 1.5 mL microtubes and stored in a freezer at −80 °C. Before each assay, a 14,000× *g* centrifugation was performed for 5 min.

### 4.4. Oxidative Damage 

To assess protein damage, the levels of carbonyl groups were determined using 2,4-dinitrophenyl hydrazine (DNPH) and measured at a wavelength of 370 nm [30]. The quantification of carbonyl levels is expressed as nmol of carbonyl per milligram of protein.

For the measurement of lipid peroxidation as an indicator, malondialdehyde (MDA) levels were determined through high-performance liquid chromatography (HPLC) using a reversed-phase column (SUPELCOSIL™ LC-18-DB HPLC column; 15 cm × 4.6 mm, 5 μm). The mobile phase consisted of a mixture of 30 mmol/L monobasic potassium phosphate (pH 3.6) and methanol (9:1, *v*/*v*) with a flow rate of 1 mL/min. Samples were injected in a volume of 25 μL, and the absorbance of the column effluent was monitored at 254 nm. The retention time of MDA under these conditions was 5.6 min [31]. MDA levels are expressed as nmol of MDA per milligram of protein. 

### 4.5. Enzymatic Activity 

The SOD activity was determined using the RanSOD kit from Randox, County Antrim, UK. The measurement of fumarase activity involved the conversion of fumarate to malate, with detection performed at 240 nm [32]. The enzymatic kinetics of glutathione peroxidase (GPx) were evaluated using the Ransel Kit from Randox, County Antrim, UK. The GST antioxidant assay was used to measure the formation of S-(2,4-dinitrophenyl)-glutathione through the enzymatic activity of GST via the conjugation of 1-chloro-2,4-dinitrobenzene (CDNB) and glutathione (GSH). The measurement of GST activity was carried out by determining the absorbance at a wavelength of 340 nm [33]. All enzyme activities are quantified as U/mg of protein.

The consumption of H_2_O_2_ was evaluated with a protocol adapted for microplates [34]. First, 30% H_2_O_2_ was diluted in 10 mL of sodium phosphate buffer (50 mmol/L, pH 7) and added into the sample to trigger the reaction, measuring the rate of H_2_O_2_ consumption via absorbance at 240 nm. H_2_O_2_ consumption was reported because there are multiple H_2_O_2_ detoxification mechanisms (mainly CAT and peroxiredoxins) and the test is not specific for any of them [35]. The activity is expressed as μmol of H_2_O_2_ consumption/min/mg of protein.

All assays were independently performed in triplicate.

### 4.6. Non-Enzymatic Antioxidants

As an index of the indirect quantification of nitric oxide levels, a variation of the Griess test was used to determine total nitrate and nitrite levels (NO_2_ and NO_3_) [34]. Indirect nitric oxide levels are expressed as nmol of NO_2_ and NO_3_/mg protein.

GSH levels were evaluated by preparing a mixture with a solution of 5,5-dithiobis (2-nitrobenzoic acid, DTNB),glutathione reductase (GR, 250 U/mL), and NADPH to convert GSSG into GSH and trigger GSH. GSSG levels were measured using treatment with 2-vinylpyridine, which reacts covalently with GSH and was measured at 412 nm [36]. The results are presented as GSSG/GSH ratio.

### 4.7. Statistical Analysis

We conducted statistical analyses using the software PAST (version 4.12) to explore the relationships between antioxidant enzyme levels, oxidative stress marker levels, and feeding guild in four bat species. Specifically, we performed one-way PERMANOVA, principal component analysis (PCA), and correlation analysis, as follows.

To test for significant differences in the mean levels of each antioxidant and oxidative stress marker among the four bat species, we conducted one-way PERMANOVA. We first checked for normality using the Shapiro–Wilk test and verified the homogeneity of variances using Levene’s test. Since these assumptions were not met (<0.05), we performed the non-parametric test one-way PERMANOVA followed by a Bonferroni-corrected PERMANOVA pairwise comparison to compare pairwise differences between the bat species.

To explore patterns of variation among the variables, we conducted PCA. Before PCA, we performed a transformation method to eliminate the effects of differences in scale, which involved subtracting the mean of each variable from its values. We then conducted PCA on the covariance matrix of the normalized variables. 

We also conducted correlation analysis to explore the relationships between the different antioxidant enzymes and oxidative stress marker levels. We calculated the Spearman correlation coefficients for pairs of variables and tested their significance using two-tailed tests. 

All statistical tests were conducted at the significance level of *p* < 0.05, and all reported *p*-values were two-tailed. We used the Bonferroni correction to adjust for multiple comparisons when applicable.

## 5. Conclusions

In conclusion, our results indicate that the levels of oxidative stress markers and the activities of antioxidant enzymes in the brain vary significantly among bat species with different feeding habitats. These findings suggest that diet and feeding habits may play an important role in the antioxidant status of the brain.

## Figures and Tables

**Figure 1 ijms-24-12162-f001:**
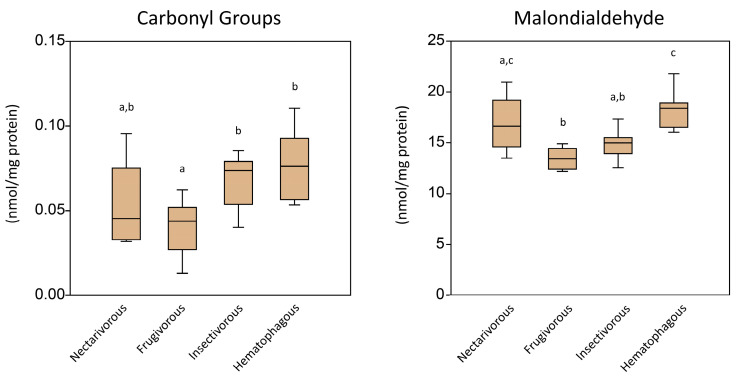
Level of oxidative damage markers measured in the brain tissue of nectarivorous, frugivorous, insectivorous, and hematophagous bats. Data are presented as the median (interquartile range). Different letters indicate statistical differences among species.

**Figure 2 ijms-24-12162-f002:**
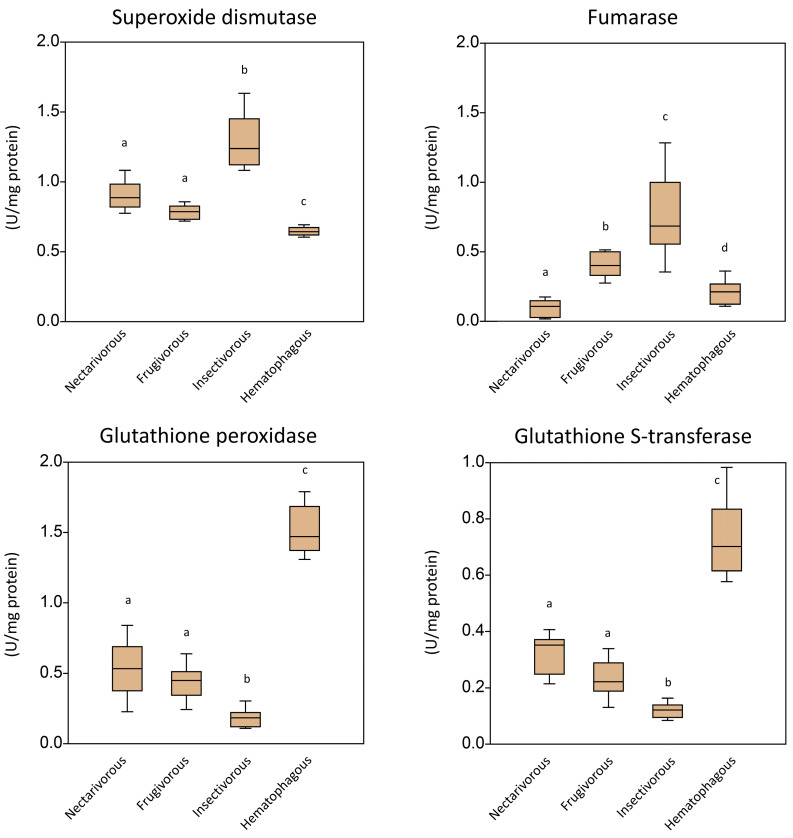
Activity of antioxidant enzymes measured in the brain of nectarivorous, frugivorous, insectivorous, and hematophagous bats. Data are presented as the median (interquartile range). Different letters indicate statistical differences among species.

**Figure 3 ijms-24-12162-f003:**
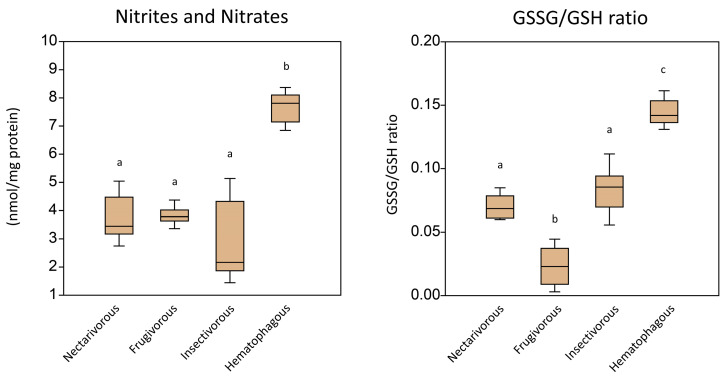
Levels of non-enzymatic antioxidants measured in the brain of nectarivorous, frugivorous, insectivorous, and hematophagous bats. Data are presented as the median (interquartile range). Different letters indicate statistical differences among species.

**Figure 4 ijms-24-12162-f004:**
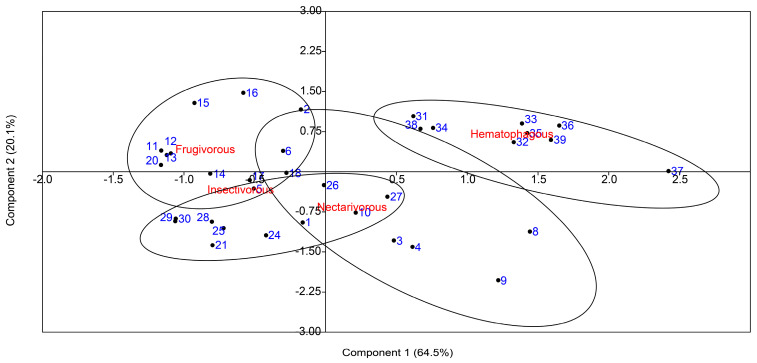
Principal component analysis of antioxidant enzymes and oxidative markers measured in the brain tissue of nectarivorous, frugivorous, insectivorous, and hematophagous bats. Numbers in blue indicate samples.

**Figure 5 ijms-24-12162-f005:**
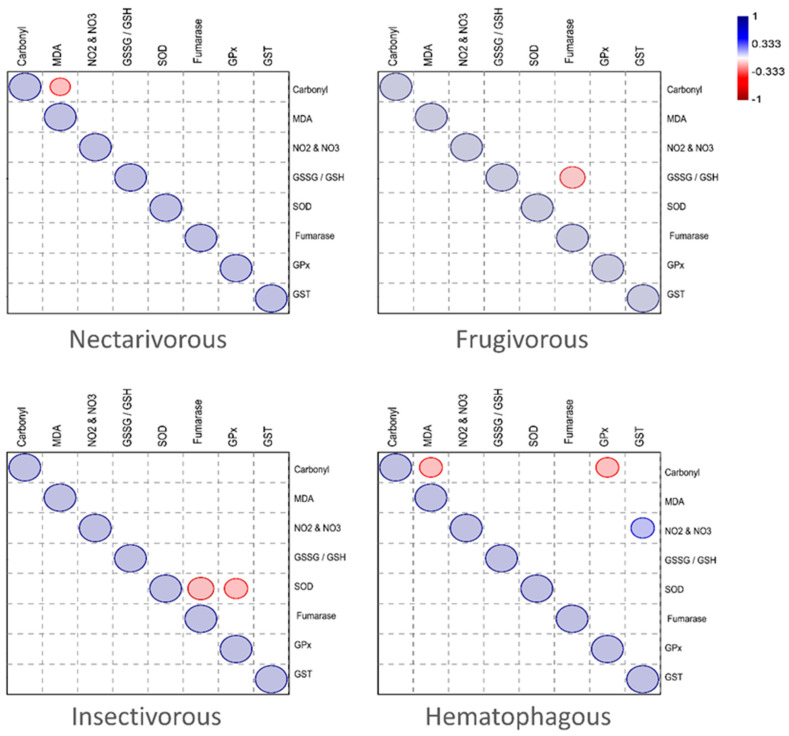
Graphical Spearman’s correlation matrix of the enzymatic and biochemical parameters in each bat species. Positive correlation (from white to blue); negative correlation (from white to red). Only significant correlations are presented (*p* < 0.05). The circle size represents the correlation coefficient.

**Figure 6 ijms-24-12162-f006:**
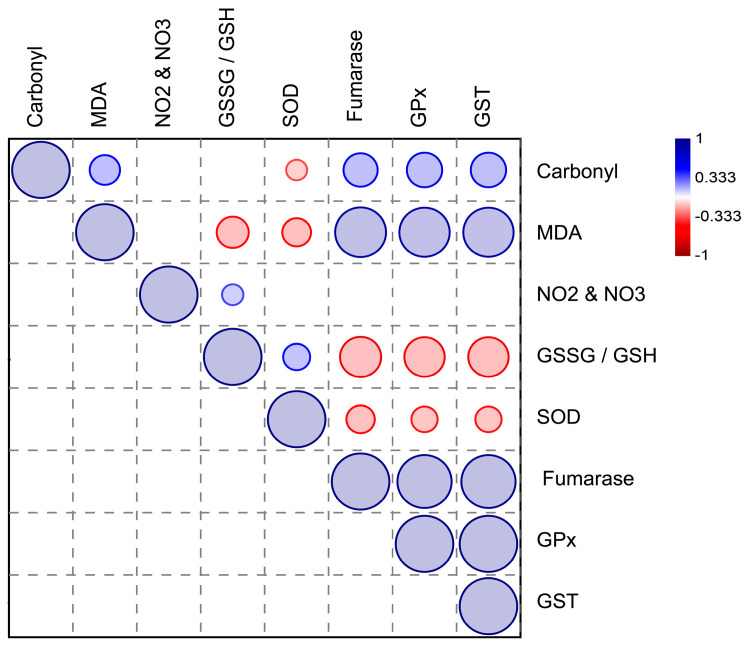
Graphical Spearman’s correlation matrix of the enzymatic and biochemical parameters measured in this study. Positive correlation (from white to blue); negative correlation (from white to red). Only significant correlations are presented (*p* < 0.05). The circle size represents the correlation coefficient.

**Figure 7 ijms-24-12162-f007:**
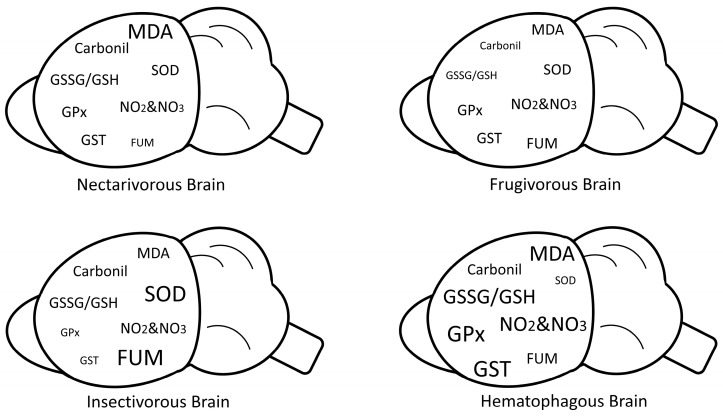
Graphical scheme summarizing the biochemical findings in each bat group. Larger letters indicate increased enzymatic activity or increased concentration of markers. Smaller letters indicate reduced enzymatic activity or reduced concentration of markers.

**Table 1 ijms-24-12162-t001:** Pairwise PERMANOVA test among the samples grouped according to bat species. Asterisks indicate statistical significance (corrected *p*-value < 0.05).

	Nectarivorous	Frugivorous	Insectivorous	Hematophagous
Nectarivorous		0.0036 *	0.0018 *	0.0006 *
Frugivorous	0.0036 *		0.0054 *	0.0006 *
Insectivorous	0.0018 *	0.0054 *		0.0006 *
Hematophagous	0.0006 *	0.0006 *	0.0006 *	

**Table 2 ijms-24-12162-t002:** The specific sites for sample collection, along with their corresponding geographical coordinates.

Bat Species	Food Group	*n*	Location (City–State)	Coordinates
*G. soricina*	Nectarivorous	10	Dom Pedro Alcântara–RS	29°24′22.35″ S 49°51′4.56″ W
*S. lilium*	Frugivorous	10	Dom Pedro Alcântara–RS	29°24′22.35″ S 49°51′4.56″
*M. molossus*	Insectivorous	10	Treviso–SC	28°30′47.52″ S 49°27′26.6″
*D. rotundus*	Hematophagous	9	Criciúma–SC	28°41′27.7″ S 49°25′50.6″

## Data Availability

Not applicable.

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
