# Peer review of "Oxidative Stress and Antioxidant Defense in the Brain of Bat Species with Different Feeding Habits"

_ijms, 2023, doi:10.3390/ijms241512162_

Round 1

Reviewer 1 Report

Int. J. Mol. Sci. 2023, 24, x. https://doi.org/10.3390/xxxxx www.mdpi.com/journal/ijms

Pabulo Henrique Rampelotto, Nikolas Raphael Oliveira Giannakos, Diego Antonio Mena Canata, FranciellyDias Pereira, Fernanda Schäfer Hackenhaar, María João Ramos Pereira and Mara Silveira Benfato

Oxidative Stress and Antioxidant Defense in the Brain of Bat Species
with Different Feeding Habits

COMMENTS FOR THE AUTHOR:

1. The presented research is an original and important for
neurobiology and biochemistry. The manuscript is included all parts which needs for thou publication: Abstract, Introduction, Materials and Methods, Results, Discussion, Conclusion, References.

2. The title clearly and precisely reflects the findings of the manuscript.

3. Abstract is it really a summary, include key findings and have an appropriate length.

4.
This study and the introduction to it demonstrate different mechanisms of antioxidant defense of the body and, above all, the brain. The authors consider the different sides of this phenomenon, these are the levels of superoxide dismutase, fumarase. On the other hand, it is the activity of glutathione peroxidase and glutathione S-transferase. On the third side is the nitric oxide system, which, in addition, is a signaling system. The authors also analyzed the products of oxidative stress - the levels of carbonyl groups and malondialdehyde. Based on the obtained results, the authors come to the assumption (conclusion) that the mechanisms of antioxidant protection depend on the diet, i.e. substances entering the body.

5. The authors described the methods briefly, but they are sufficient for reproduction by other researchers. There is no need for additional materials.

6. The general logic of the results is correct, the pictures suggested and located strictly in accordance with the sсheme of section. In my opinion, additional experiments are not required.

I didn't find section 3.2.

7. The discussions take a detailed look at the mechanisms of antioxidant defense using the example of animals that prefer different diets, I would say strikingly different. The results obtained show that these mechanisms differ significantly in animals with different diets. I just do not agree with the strongly straightforward statement of the authors “These results suggest that the antioxidant defense mechanisms in the brain of bats are influenced by their feeding habits”, one cannot exclude the inverse dependence of a certain development of different antioxidant defense mechanisms in the evolutionary process.

8. The figures correspond to the article’s structure; legends to him explain the drawings. The citation is appropriate, the included the basic publications on the topic.

9. Final comments.

The manuscript is fully consistent to the stated theme. I recommend for publication without additions and corrections.

Khalil L. Gainutdinov – Dr.Sci.Biol., Professor, professor of Department of Human and Animals Physiology, Institute of Fundamental Medicine and Biology of Kazan Federal University, Kremlevskaya st., 18, Kazan, 420008; [email protected]

Author Response

We thank the reviewers for the excellent critical comments. 

Reviewer 2 Report

The present research work is aimed to investigate the oxidative brain damage and antioxidant status of four bat species (39 wildlife captured animals) with different feeding habits (insectivorous, frugivorous, nectarivorous and hematophagus) with the final aim to ‘gain insights into the protective mechanisms against oxidative stress and the interspecific variation’. The results confirm the hypothesis of such differences, however a critical issue is to which extend the differences can be only atributed to diets and feeding habits or to other interespecific differences in other basic functions.

Introduction and whole manuscript. The ecological rationale for the interest of the bats is poorly explained (or really does not exists, as it is not the purpose, but the bats are just a tool), so the article is focused on the impact of diets and feeding habits on brain oxidative status and defenses. The rational provided in the last paragraph of the introduction seems to indicate that it is a translational approach for dietary strategies in humans. If this is the case, cues of such translation should be given, i.e. equivalence of each feeding behavior to that in humans.

Lines 501-57. As mentioned, it is not clear that the findings on differences of brain oxidative status and defenses can be attributed or are due to the feeding habits, as they could be also due or influenced by differences in other basic physiological habits (sleeping behavior, for instance) or metabolic/energy patterns. In fact, on  line 222, the authors refer to a range of metabolic adaptations. At least, these aspects should be more discussed and/or clues in favour/against to be discarted should be given.

Methods. Since wild animals are used, how the authors monitored the impact of stress of capture, and the cronobiological impact on results of the wide range of that procedure of ‘recruitment’ (summer to winter).

Results. Presentation of data should be in agreement with the order of the categories given at the beginning (insectivorous, frugivorous, nectarivorous and hematophagus) or if the authors (as it seems) have chosen a visual presentation that shows the progression patterns (i.e. Glutathione peroxisade and S-transferase) to emphasize the findings, then the initial argument should be similar. Still, the principal component analysis (figure 4) yields a ierarquic structure which is different: frugivorous, insectivorous, nectarivorous, and hematophagous.

Figure 5. Should be of interest to also see the matrix for each specie, so it can illustrate the ‘interspecific variation’ that authors refer to.

In this respect, the discussion should also provide a summary picture of each specie (a translation of figure 5), so have a clear rationale for each ‘dietary and feeding habits’ as the final purpose is to find a translation into different diets and feeding habits in humans.

Author Response

The present research work is aimed to investigate the oxidative brain damage and antioxidant status of four bat species (39 wildlife captured animals) with different feeding habits (insectivorous, frugivorous, nectarivorous and hematophagus) with the final aim to ‘gain insights into the protective mechanisms against oxidative stress and the interspecific variation’. The results confirm the hypothesis of such differences, however a critical issue is to which extend the differences can be only atributed to diets and feeding habits or to other interespecific differences in other basic functions.

Reply: We thank the reviewers for the excellent critical comments. We provide below a point-by-point reply to them and also addressed them in the revised version of the manuscript (highlighted in red).

Introduction and whole manuscript. The ecological rationale for the interest of the bats is poorly explained (or really does not exists, as it is not the purpose, but the bats are just a tool), so the article is focused on the impact of diets and feeding habits on brain oxidative status and defenses. The rational provided in the last paragraph of the introduction seems to indicate that it is a translational approach for dietary strategies in humans. If this is the case, cues of such translation should be given, i.e. equivalence of each feeding behavior to that in humans.

Reply: We agree that the ecological rationale of the study was poorly explained, so we have rewritten the last paragraph of the introduction to make it clearer. 

Lines 501-57. As mentioned, it is not clear that the findings on differences of brain oxidative status and defenses can be attributed or are due to the feeding habits, as they could be also due or influenced by differences in other basic physiological habits (sleeping behavior, for instance) or metabolic/energy patterns. In fact, on  line 222, the authors refer to a range of metabolic adaptations. At least, these aspects should be more discussed and/or clues in favour/against to be discarted should be given.

Reply: Excellent point raised by the reviewer, which gives us the opportunity to better address it in the manuscript.  All the collected animal species in our study exhibit nocturnal habits and follow similar circadian cycles. They share similar foraging times, including feeding and flight activities. Moreover, we observed that the entry and exit times of the caves were practically identical for all species. Notably, none of these four species engage in hibernation or torpor. So, based on our observations, we believe that the differences in these habits among the species are not significant enough to account for the variations we observed and analyzed in terms of redox metabolism. Instead, we attribute these differences to variations in the diet of these species. While we briefly pointed out at the end of the discussion that “we were able to control major factors that are common interferences in wildlife studies”, the reviewer's comment made us realize that this issue should be better explained in the manuscript, so we added a new paragraph at the beginning of the discussion to address it.

Methods. Since wild animals are used, how the authors monitored the impact of stress of capture, and the cronobiological impact on results of the wide range of that procedure of ‘recruitment’ (summer to winter).

Reply: The stress associated with the capture of the animals, including the time spent in the net, handling of the individuals, and euthanasia procedures, was consistent among all four species. These activities were carried out by the same individuals, ensuring uniformity in the stressors applied. Consequently, any biases that could have arisen from these procedures would have been equivalent for all species, minimizing the potential for significant intraspecies differences. This issue was also addressed in the second paragraph of the discussion.

Results. Presentation of data should be in agreement with the order of the categories given at the beginning (insectivorous, frugivorous, nectarivorous and hematophagus) or if the authors (as it seems) have chosen a visual presentation that shows the progression patterns (i.e. Glutathione peroxisade and S-transferase) to emphasize the findings, then the initial argument should be similar. Still, the principal component analysis (figure 4) yields a ierarquic structure which is different: frugivorous, insectivorous, nectarivorous, and hematophagous.

Reply: In fact, the presentation of data was always presented in the same order, from methods to the discussion (nectarivorous, frugivorous, insectivorous, and hematophagus). In addition, it is the same pattern we used in other articles. For these reasons, we would rather keep the standard.

Figure 5. Should be of interest to also see the matrix for each specie, so it can illustrate the ‘interspecific variation’ that authors refer to.

Reply: Following this suggestion, we included the matrix for each species (Figure 5), and describe it in results (lines 223-228) and discussion (lines 370-404). The overall correlation matrix is now Figure 6.

In this respect, the discussion should also provide a summary picture of each specie (a translation of figure 5), so have a clear rationale for each ‘dietary and feeding habits’ as the final purpose is to find a translation into different diets and feeding habits in humans.

Reply: Following this suggestion, we included a summary picture of our biochemical findings in each species. Please see Figure 7.

Round 2

Reviewer 2 Report

The authors have provided a detailed point-by-point answer to all comments and questions, and have done an accurated work to add paragraphs that address and perfectly solve the issues raised. In some other cases, they have followed the standard, in order to be consistent with their previous reports.
The authors have also succesfuly adressed new questions and analysis, that are very appreciated, as the matrix for each bat specie perfectly illustrates specific correlation patterns and commonalities. Also, Figure 7 and discussion per species, are new parts that have been done with accuracy and add value/extend their findings.

Please, just check a minor point: Why when considering the whole sample, the correlation analysis between MDA and carbonyl or that between GPx and carbonyl are positive (blue) while when analyzed per species, the species that show such a correlation  (nectarivorous and hematophagous, in the case of MDA; hematophagus, in the case of GPx)  have it negative (red).

Thanks

Author Response

We believe this suggests that in these specific species, the relationship between MDA and carbonyl, as well as GPx and carbonyl, works differently from the overall sample. It could be that the antioxidant defense mechanisms in these species respond differently to oxidative stress, leading to variations in the relationship between MDA, GPx, and carbonyl. Meaning, the positive correlation between MDA, GPx, and carbonyl in the whole sample may reflect a general relationship between oxidative stress and antioxidant defenses. However, the negative correlations observed when analyzed per species indicate that specific factors from different species could influence these relationships differently.